# Air-Coupled Ultrasound Sealing Integrity Inspection Using Leaky Lamb Waves in a Simplified Model of a Lithium-Ion Pouch Battery: Feasibility Study

**DOI:** 10.3390/s22176718

**Published:** 2022-09-05

**Authors:** Hyunwoo Cho, Eunwoo Kil, Jihun Jang, Jinbum Kang, Ilseob Song, Yangmo Yoo

**Affiliations:** 1Department of Electronic Engineering, Sogang University, Seoul 04107, Korea; 2Medical Solutions Institute, Sogang University, Seoul 04107, Korea; 3Department of Bioengineering, University of Washington, Seattle, WA 98105, USA; 4Department of Biomedical Engineering, Sogang University, Seoul 04107, Korea

**Keywords:** non-destructive testing, lithium-ion battery, aluminum pouch film, air-coupled ultrasound, leaky Lamb wave

## Abstract

Inspecting the sealing integrity of lead tabs is an important means of ensuring the reliability and safety of pouch-type lithium-ion (Li-ion) batteries with a thin multi-layered aluminum (Al) laminated film. This paper presents a new air-coupled ultrasonic non-destructive testing (NDT) inspection method based on leaky Lamb wave transmission; and reception for evaluating the sealing integrity between the lead tab and the Al pouch film. The proposed method uses the critical incidence angle between the air and the layer with the fastest Lamb wave velocity to maximize the signal-to-noise ratio in the through-transmission mode. To determine the critical incidence angle, phantom experiments with two test pieces (i.e., an Al tab and an Al tab sealed with an Al pouch film) are conducted. In addition, 2D scans are performed at various incidence angles for an inhouse pouch-type Li-ion battery with a 1-mm-wide foreign material inserted as a defect. At the critical incidence angle (i.e., 22°), the proposed air-coupled ultrasonic NDT method in through-transmission mode successfully identifies the shape and location of the defect through c-scan image reconstruction. These preliminary results indicate that the proposed air-coupled ultrasonic NDT method with leaky Lamb waves can be used to inspect the sealing integrity of Li-ion pouch batteries in dry test conditions.

## 1. Introduction

As lithium-ion (Li-ion) batteries become more popular in the electric vehicle and energy storage system markets, the need to inspect their reliability and safety is gradually increasing. Accordingly, there is an increasing demand for aluminum (Al) pouch film for manufacturing Li-ion pouch batteries; thus, a method of inspecting the thin multilayer Al film structure is required. In particular, the sealing integrity between the lead tabs of Li-ion batteries and the Al pouch film is important; this is because it directly affects the stability and life of the Li-ion battery [1,2]. Long-term stability is only possible when the sealing is guaranteed to prevent electrolytes from leaking; or moisture from infiltrating into the Li-ion battery and shortening its lifespan. To ensure the safety and stability of Li-ion batteries, the need for the reliable inspection of sealing integrity in an Al pouch film has grown.

Various testing methods have been proposed for inspecting the sealing integrity of an Al pouch film; e.g., the gas leak test, vacuum decay test, bubble emission test, and dye penetration test [3]. These methods are destructive, operator-dependent, and have limited resolution; hence, they are not appropriate for inspecting the sealing integrity of pouch-type Li-ion batteries [4]. Alternatively, ultrasonic inspection, which is commonly used in various non-destructive testing (NDT) applications, can be applied to Li-ion pouch batteries. An ultrasonic NDT method based on a backscattered amplitude integral mode was previously proposed for evaluating defects in flexible plastic materials and Al retort pouches; which have similar structures to Al pouch films [5,6,7]. This ultrasonic NDT method was applied to monitoring the physiochemical nature of the lithium-ion battery [8,9,10].

Although this ultrasonic NDT method can reliably detect channel defects in Al pouch films, it cannot be applied to the inspection of Li-ion pouch batteries because it can only be performed in immersion conditions. For dry testing, X-ray inspection can be applied; however, this approach does not have sufficient resolution and sensitivity to detect air channels and foreign bodies between the lead tabs and Al pouch films. Therefore, a new reliable NDT method is needed to inspect the sealing integrity of Li-ion pouch batteries under dry conditions.

Air-coupled ultrasonic NDT may be a suitable solution for inspecting the Li-ion battery pouch sealing integrity in non-immersion conditions. However, applying air-coupled ultrasonic NDT in this situation using the conventional through-transmission mode is challenging because of the enormous signal loss caused by the very high acoustic impedance mismatch between air and solid materials. In other words, when ultrasound waves progress from air to solid materials, most of the acoustic signal is reflected. Thus, in the conventional through-transmission mode, the amplitude of the signal obtained through test pieces (e.g., lead tabs) becomes very small; this makes it difficult to detect the structure and presence of defects. For this reason, conventional ultrasonic NDT is not suitable for inspecting the structure and defects of objects in air-coupled conditions.

Ultrasonic NDT using Lamb waves has been widely investigated because it can amplify the transmission energy, especially in thin plates or film structures [11,12,13,14]. Lamb waves occur in thin plates (e.g., lead tabs) or multilayered structures; generated by the phenomenon whereby transverse/longitudinal waves resonate towards the top and bottom inside a structure and proceed in the in-plane direction of the plate [13,14]. These waves are generated in various modes depending on the physical properties of the test piece and the frequency of the transmitted ultrasound wave. Accordingly, Lamb wave velocities and critical incidence angles can be determined based on Snell’s law [12,13,14].

To obtain relatively high signal-to-noise ratios (SNRs) using Lamb waves, it is important to determine an appropriate incidence angle and mode according to the structure of the test piece. Cawley et al. [15] suggested the appropriate Lamb wave mode, as well as its excitation and reception, for evaluating large structures (e.g., steel plate). Castaings et al. [13] conducted a numerical study on the interaction between Lamb waves and defects. By examining the notches of steel plates, the optimal conditions for detecting small defects, such as the optimal incidence mode and angle for transmission energy amplification, were successfully determined. Bar et al. [14] experimentally measured the incidence angle that produced the maximum transmission energy in test pieces made of steel, Al, carbon-fiber-reinforced composite, and honeycomb sandwich panels. As the Lamb waves proceed through the plate structure, energy leaks to the surrounding media (e.g., air); this is a phenomenon known as leaky Lamb waves [16]. The energy leakage generated in this way takes the form of an oblique pressure wave, and the angle of leakage pressure diverges to the critical angle of the loading material [16,17]. Recently, Koller et al. [18] conducted a thorough study where Lamb wave modes on Li-ion batteries were determined with piezoelectric transducers. In addition, ultrasonic NDT inspection methods have been investigated for evaluating the electrodes or cell structure of Li-ion batteries by detecting Lamb waves [19,20]. While these studies showed promising results for evaluating battery structures, the sealing inspection method was not thoroughly investigated using the Lamb wave in non-immersion settings.

In this paper, a new air-coupled NDT inspection method based on leaky Lamb wave transmission and reception is presented for evaluating the sealing integrity between lead tabs and the Al pouch film. First, the optimal ultrasound incidence angle for Li-ion pouch battery inspection was determined using the appropriate mode and velocity of the Lamb waves. Second, this result was experimentally verified through a phantom experiment in which two test pieces are evaluated; i.e., an Al tab and an Al tab sealed with an Al pouch film. Lastly, using the incidence angle that allows the maximum Lamb wave leakage signal to be measured, the shape and structure of a 1-mm-wide foreign material (i.e., paper) was successfully identified in 2D scanning of an inhouse pouch-type Li-ion battery model.

## 2. Materials and Methods

Figure 1a,b illustrate the generation, propagation, and leakage of Lamb waves in a single plate (i.e., an Al lead tab) and a multi-layered structure (i.e., an Al lead tab sealed with an Al pouch film).

As shown in Figure 1a, the Lamb wave generated from Material 1 by an ultrasound wave with an incidence angle of $\theta_{inc}$ propagates along the plate (i.e., an Al lead tab); leading to energy leakage to the loading fluid (i.e., air). This energy leakage takes the form of an oblique pressure wave and proceeds based on Snell’s law according to the properties of the adjacent media. The angle of the leakage pressure wave angle, i.e., $\theta_{leakage}$, is determined by
(1)$$\theta_{leakage}={sin}^{-1}\left(\frac{V_{air}}{V_{Lamb}}\right),$$
where $V_{air}$ and $V_{Lamb}$ denote the wave velocity in air (which is the adjacent medium), and the Lamb wave velocity in the surface of the multi-layered structure (where the leakage occurs), respectively. As can be seen in Figure 1b, similar leaky Lamb waves occur in multi-layered structures. The Lamb waves generated in the multi-layered structure proceed in different layers before leaking, and the angle of the oblique pressure wave is determined by the properties of the layers passed through [22]. Finally, the leaky Lamb waves pass through the whole multi-layered structure and proceed to the loading fluid (i.e., air).

Figure 1c shows the phase velocity dispersion curve for an Al plate, which is used as the lead tab in Li-ion pouch batteries. Depending on the value of the f-d product obtained by multiplying the ultrasound frequency and the Al layer thickness, the velocity, critical angle, and wave number of each Lamb wave can be expressed as a dispersion curve [11,12,13,14,15,16]. Lamb waves have dispersive characteristics; thus, there are many modes, and an appropriate mode must be selected according to the application. The symmetric mode (S mode) mainly generates an in-plane surface displacement (especially S0) and has relatively nondispersive characteristics. In the case of the anti-symmetric mode (A mode), its surface motion largely occurs in the out-of-plane direction; i.e., relatively dispersive characteristics [15]. In particular, it is easy to detect leakage from the A0 mode because the dominant surface motion is generally in the out-of-plane direction.

The thickness of each layer in the multi-layered Al pouch film used in Li-ion pouch batteries is typically less than 40 μm, and that of the Al lead tab is less than 200 μm. As the center frequency of the incident ultrasound wave does not exceed 1 MHz, the *f-d* value is less than 0.2 (MHz × mm). As indicated in Figure 1c, the mode of the Lamb waves generated when an Al plate is used as the lead tab is limited to the S0 and A0 modes. Therefore, in the proposed method, the leakage of Lamb waves through the A0 mode is detected.

The theoretical value of the critical incidence angle given in Equation (1) maximizes the ultrasound transmission energy passing through the material. This value was experimentally determined for a multi-layered test piece; i.e., an Al tab sealed with an Al pouch film. In the multi-layered structure, there is a layer (hereinafter referred to as the critical layer) that exhibits the largest critical incidence angle due to the fastest Lamb wave speed and largest volume fraction. In the case of Li-ion pouch batteries, the Lamb wave velocity of the A0 mode at the Al lead tab layer is the fastest at the same frequency. Since the behavior of the Lamb wave through the air cavity (i.e., the defect in the test pieces) is different from the original multi-layered structure, the air cavity can be detected by the leakage angle of the Lamb wave.

For the experiment, a system of data acquisition and control operation was utilized similarly to the previous study for real-time monitoring and control operations [23]. Considering the capability of being adopted on various control components, the host of the controller is selected as a PC in our study. Figure 2a shows the experimental setup used to determine the critical angle at the highest SNR of the signal passing through the test piece. The 1-Vpp, 700-kHz, 3-cycle sine wave from the function generator (AFG 3252, Tektronix, Beaverton, OR, USA) is amplified by the RF Amplifier (350L, Electronics & Innovation, Rochester, NY, USA); this results in approximately 315 Vpp derived to the 700 kHz air-coupled transducer with a focal depth of 50 mm (NCT700-D2 5-P50, Ultran Group, New York, NY, USA). The received signal is amplified by 60 dB through the ultrasound receiver (5800PR, Olympus, Tokyo, Japan) and transmitted to the PC.

To evaluate the proposed method, a simplified model of the Li-ion pouch battery was made with an Al lead tab and an Al pouch film. Materials for the layers are 99.0% pure Al for the Al tab, and Al laminated film (Al-plastic film) was used as the Al pouch film. The Al laminated film is composed with polyamide, Al foil, polypropylene, and adhesive materials between each layer of the Al laminated film. Figure 2b shows two test pieces; i.e., the Al lead tab (~200 μm) and the Al lead tab sealed with the Al pouch film (~150 μm), using adhesive polyimide (PI) film (~40 μm). As shown in Figure 2c, the inhouse Li-ion pouch battery defect model was made by attaching double-sided adhesive PI film to the Al lead; Table A foreign material (i.e., paper varying from a width of 0.8–1.3 mm and a thickness of 0.07 mm) was inserted between the attached PI film and the Al pouch film, and heat-sealing was performed. The foreign material was used to generate ~1-mm-wide air pockets between the Al lead tab and the Al pouch film. The inhouse Li-ion pouch battery defect model contains an Al sheet layer (<40 μm) inside the Al pouch film. However, the Al plate, i.e., the Al lead tab, is the critical layer; this is because Lamb waves with the A0 mode velocity (1115 m/s) in the Al plate are faster than those in the Al sheet layer (530 m/s). Therefore, as the three test pieces have the same critical layer, a similar critical angle should be measured.

## 3. Results and Discussion

Figure 3a shows the normalized amplitude of the signal passing through the Al lead tab and the Al lead tab sealed with the Al pouch film; whose structures are illustrated in Figure 2b, with respect to the angle formed by the test piece and the air-coupled transducer. For each test piece, 50 scans were performed and the results were averaged. The angle resolution was 0.03°. The peak values were recorded at 20.87 ± 0.37° for the Al lead tab and 22.11 ± 1.06° for the Al lead tab sealed with the Al pouch film. The ideal critical angle of the Al lead tab obtained from Equation (1) is 18.02° when the speed of sound in the air is 345 m/s and the Lamb wave velocity with the A0 mode is 1115 m/s. Thus, the measured value from the test piece experiment was around 20.87°. The difference between the ideal and measured values is around 2.85°; this is possibly due to local incidence angle changes caused by local bending in the experiment. Within the range of different values, the normalized amplitude remained larger than the half of the peak value; i.e., −6 dB, which means highly distinguishable from background noise. The critical angle for the Al lead tab sealed with the Al pouch film was 22.11°; this value was used to demonstrate the detectability of defects in the Li-ion pouch battery defect model.

In addition, 2D scanning was performed with a 2-axis linear stepping motor stage (SGSP26-100, SIGMA KOKI, Tokyo, Japan) by mechanically sweeping the transmission and reception air-coupled ultrasonic transducers. Figure 3b shows the 2D scan result of the test piece with air-pocket defects in Figure 2c at 0°, 22°, and 40°. The scan size was 90 mm × 25 mm; and the scanning steps in the x- and y-directions were 0.02 mm and 0.5 mm, respectively. Gamma correction (gamma = 5) was applied to the measured data by adjusting the contrast, allowing the defect to be clearly depicted. As shown in Figure 3b, the defect, i.e., “MICS,” was clearly observed at a critical incidence angle of 22°; whereas, the defect cannot be detected in the 0° and 40° scans because the noise-level signals are dominant. Note that defects with a width of ~1 mm appear to measure ~2 mm because the air-coupled ultrasonic transducer has a 1.5-mm beam width at the focal point (50 mm). As shown in Figure 3b, the signal passes through the sealing part without defects, i.e., MICS; it is then lost in the part containing foreign substances, i.e., air pockets. Considering the results shown in Figure 3b, the proposed air-coupled NDT evaluation method has the potential to detect foreign materials or air pockets between the Al lead tab and the Al pouch film. However, further studies are needed to clarify the subtle effects of layers other than the critical layer on the critical angle in multi-layered structures; i.e., Li-ion pouch batteries. In addition, the sensitivity of the incidence angle can be reduced at higher frequency–thickness products by using a higher ultrasound frequency. Even smaller defects could be accurately detected using an air-coupled transducer with better beam resolution. The proposed method with the detection of leaky Lamb waves showed the improved spatial and contrast resolution compared to the air-coupled ultrasonic NDT method with the through-transmission technique in the previous study [24].

## 4. Conclusions

This paper has investigated sealing integrity inspection on a simplified model of Li-ion pouch batteries using air-coupled ultrasound NDT. The concept of the critical layer was introduced to determine the critical angle that maximizes the magnitude of Lamb wave generation and leakage signal detection in a thin multi-layered structure (thickness of each layer ~30–200 μm). In the multi-layered structure, it was experimentally verified that the overall critical angle can be approximated by the critical angle formed by the surrounding fluid (i.e., air) and the critical layer with the fastest Lamb wave velocity. Seal integrity inspection was performed with a relatively small signal of about 300 Vpp; in addition, 1-mm-wide defects were successfully detected at the critical angle obtained from the experiment. These preliminary results indicate that the proposed air-coupled NDT method with leaky Lamb waves can be used to evaluate the sealing integrity of Li-ion pouch batteries in non-immersion conditions after further validation. Further studies about the faster scanning method to quickly find critical incidence angles and general solutions for the non-immersion NDT method with more complex structures of various pouch batteries will be conducted.

## Figures and Tables

**Figure 1 sensors-22-06718-f001:**
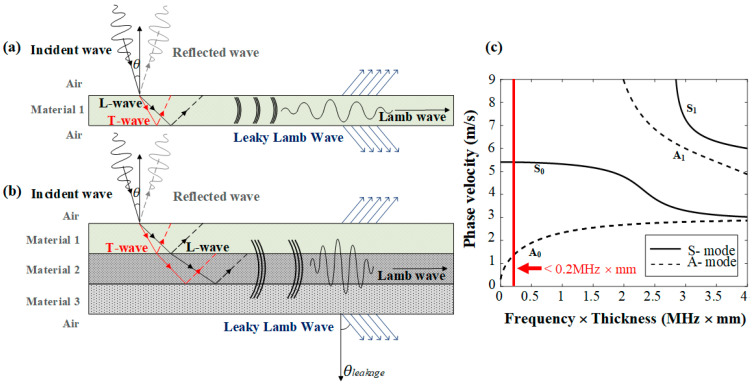
Illustration of Lamb wave generation, propagation, and leakage in (**a**) a single plate and (**b**) a multi-layered structure. (**c**) The phase velocity dispersion curve in Al [21].

**Figure 2 sensors-22-06718-f002:**
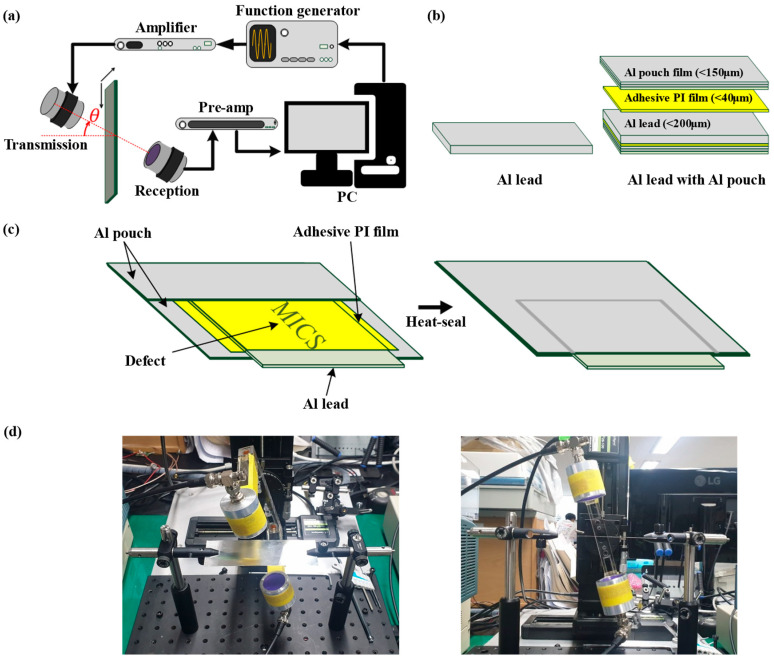
(**a**) The experimental setup of the air-coupled ultrasonic NDT method. (**b**) The test pieces used in the experiment. From the left, the Al lead tab, and the Al lead tab sealed with the Al pouch film and adhesive polyimide (PI) film. These structures are all symmetrical. (**c**) The inhouse Li-ion pouch battery defect model in which a 1-mm piece of paper was inserted as a foreign material. (**d**) Photographs of the experimental setup.

**Figure 3 sensors-22-06718-f003:**
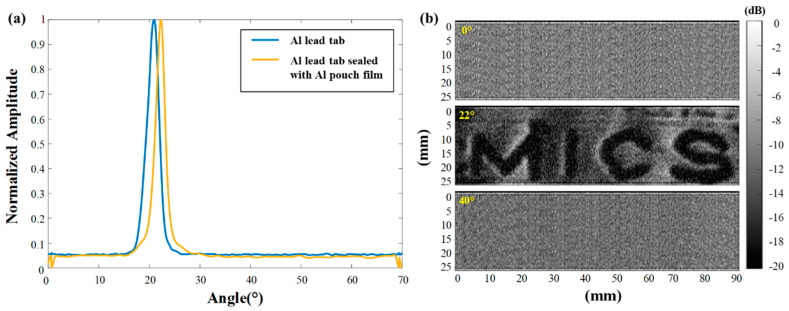
(**a**) Angle experiment results of the single Al lead tab and the Al lead tab sealed with the Al pouch film. All the test pieces have peak values of 20~23°. (**b**) A 2D scan result of the inhouse Li-ion pouch battery defect model with foreign defect material. The incident angle θ was successively set to 0°, 22°, and 40°.

## Data Availability

The data that support the findings of this study are available from the corresponding author upon reasonable request.

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
