# Peer review of "Air-Coupled Ultrasound Sealing Integrity Inspection Using Leaky Lamb Waves in a Simplified Model of a Lithium-Ion Pouch Battery: Feasibility Study"

_sensors, 2022, doi:10.3390/s22176718_

Round 1

Reviewer 1 Report

In the paper, the authors are dealing with an emerging topic of non-destructive evaluation of Li-ion battery condition state with ultrasonic techniques. With the proposed ultrasonic air-coupled testing methodology, air cavities have been visualized with high precision in the experimental specimen intending to simulate real-case lithium-ion pouch battery. In general, the paper is within the Scope of Sensors journal and the obtained results might of high potential interest for a broad audience. However, certain improvements could be introduced to the paper before its acceptance.

Some particular questions and comments are as follows:

1) Since in the manuscript no particular lithium-ion pouch battery was studied but rather its simplified model was considered, the title of the paper should be revised to avoid any misunderstanding by the readers.

2) Abstract could be made slightly shorter by removing some particular details regarding the research output, i.e., the values of critical angles, information about the transducer central frequency, etc.

3) In the Introduction, the authors might like to provide more thorough revision of current advancements of ultrasonic based testing of Li-ion batteries. For instance, papers https://doi.org/10.3390/electronics8070751, https://doi.org/10.1002/aenm.202000806, https://doi.org/10.1016/j.joule.2020.07.014 could be cited.

4) Line 82: It is mentioned that "energy leaks to the surrounding media (e.g., Al pouch film)". Typically, when considering leaky waves it is supposed that the elastic waveguide of finite thickness is immersed into some acoustic medium (as it is considered in Ref. 13, for instance). Therefore, employing the term "leaky Lamb waves" for the case when the middle layer is surrounded by thin film is probably not the best practice.

5) The authors might like to provide some additional information about the materials used in experiments, by instance, their densities and elastic properties. Moreover, the procedure which was used to manufacture the evaluated experimental sample could be described in more details.

6) Figure 1: It is not clear why for a multilayered structure, Lamb waves are located only in the middle layer while in two external layers leaky waves are shown. From the physical point of view, for such structure, elastic guided waves are propagating along the whole laminate waveguide without such separation, while wave energy leaks to the surrounding acoustic half-spaces.

7) Theoretical derivations with Eq. 1 were performed only for a single layer waveguide. It might be plausible, if authors evaluate theoretical dispersion curves for a considered multilayered structure and calculate such critical angle for it as well.

8) Line 139: "Assuming that the wave velocity in each layer is constant..." Why such assumption is valid?

9) Line 157: What was the thickness of this paper?

10) Line 178: Why such bending occurred? Was there any attempt to avoid it?

11) Line 185: How this 2D scanning was performed? What type of equipment was used for it?

12) Line 187: What was the reason for such variation of step size for different directions?

13) In Conclusion section, the authors should be cautious with such a generalization. In fact, the obtained results and corresponding conclusions could be suitable only for the particular structure considered.

Author Response

We would like to thank the reviewers for her/his valuable comments. We are resubmitting the revised manuscript and our responses to the comments and suggestions made by each reviewer. Every effort has been made to address the reviewers’ concerns, and any changes have been highlighted in the revised manuscript.

Reviewer 2 Report

Dear Authors,

The topic of the article is interesting and contains novelties. However, I believe that the article can be published if some corrections are made. Suggested corrections are listed below.

At the end of the introduction part, what was done in the study should be written step by step.

The novelty of the study and its difference from other studies should be revealed.

For such a study, the literature should be expanded using larger and more up-to-date sources. For example “Determination of Lamb Wave Modes on Lithium-Ion Batteries Using Piezoelectric Transducers”

It would be beneficial to evaluate the success of the method expressed in the article according to the literature.

It will be useful to interpret the success of this method technically in a strong way.

The applicability of the method used should be demonstrated.

In the introduction, the importance of the study and its scientific contribution should be written in clear terms.

The experimental setup set up in Figure 2 should be explained to the reader in more detail. In particular, technical details should be given.

If there are photographs of the experimental setup, they should be attached.

You can refer to the article given below regarding the setup and explanation of the experimental setups. Design and Implementation of Real-Time Monitoring and Control System Supported with IOS/Android Application for Industrial Furnaces.

It would be useful to expand the conclusion and discussion section.

Yours sincerley

Author Response

(The authors gave the same response as above.)

Round 2

Reviewer 1 Report

The authors have carefully responded to all the comments. The paper could be accepted.